# The Development of a Uniform Alginate-Based Coating for Cantaloupe and Strawberries and the Characterization of Water Barrier Properties

**DOI:** 10.3390/foods8060203

**Published:** 2019-06-11

**Authors:** Tugce Senturk Parreidt, Martina Lindner, Isabell Rothkopf, Markus Schmid, Kajetan Müller

**Affiliations:** 1Technical University of Munich, TUM School of Life Sciences Weihenstephan, Weihenstephaner Steig 22, 85354 Freising, Germany; martina.lindner@ivv.fraunhofer.de (M.L.); isabell.rothkopf@ivv.fraunhofer.de (I.R.); 2Fraunhofer Institute for Process Engineering and Packaging IVV, Giggenhauser Straße 35, 85354 Freising, Germany; kajetan.mueller@hs-kempten.de; 3Albstadt-Sigmaringen University, Faculty of Life Sciences, Anton-Günther-Str. 51, 72488 Sigmaringen, Germany; schmid@hs-albsig.de; 4Faculty of Mechanical Engineering, University of Applied Science Kempten, Bahnhofstraße 61, 87435 Kempten, Germany

**Keywords:** edible coating, edible film, sodium alginate, fruits, coating uniformity, dipping, water loss, water activity, water vapor resistance, water sorption

## Abstract

Water loss, gain or transfer results in a decline in the overall quality of food. The aim of this study was to form a uniform layer of sodium alginate-based edible coating (1.25% sodium alginate, 2% glycerol, 0.2% sunflower oil, 1% span 80, 0.2% tween 80, (*w*/*w*)) and investigate the effects on the water barrier characteristics of fresh-cut cantaloupe and strawberries. To this end, a uniform and continuous edible film formation was achieved (0.187 ± 0.076 mm and 0.235 ± 0.077 mm for cantaloupe and strawberries, respectively) with an additional immersion step into a calcium solution at the very beginning of the coating process. The coating application was effective in significantly reducing the water loss (%) of the cantaloupe pieces. However, no significant effect was observed in water vapor resistance results and weight change measurements in a climate chamber (80%→60% relative humidity (RH) at 10 °C). External packaging conditions (i.e., closed, perforated, and open) were not significantly effective on water activity (a_w_) values of cantaloupe, but were effective for strawberry values. In general, the coating application promoted the water loss of strawberry samples. Additionally, the water vapor transmission rate of stand-alone films was determined (2131 g·100 µm/(m^2^·d·bar) under constant environmental conditions (23 °C, 100%→50% RH) due to the ability to also evaluate the efficacy in ideal conditions.

## 1. Introduction

Fruits and vegetables supply dietary fibers, vitamins, minerals, and phytochemicals that have functions, such as phytoestrogens, antioxidants, anti-inflammatory agents, etc. [1,2]. Fruit and vegetable consumption must be increased to optimize nutrition uptake, promote health due to the strong link between fruit and vegetable intake, and decrease risk of cancer, heart disease, stroke, cataracts, diverticulosis, chronic obstructive pulmonary disease, and hypertension [3]. Due to the increasing demand of consumers for fresh, healthy, additive-free, and ready food products with reduced preparation time, the fresh-cut fruits and vegetables market has grown perpetually in the European Union (EU) [4,5]. According to the 2015 market research report presented by Euromonitor International [6], a 19% per capita volume growth for fresh-cut fruit in Western Europe was recorded. Fresh-cut produce are fruits and vegetables that have been cleaned, peeled, cored, chopped, sliced, diced, and packaged. As a result of these physical processes, fresh-cut products are more perishable and susceptible to physiological–biochemical changes and microbial degradation [7,8]. On top of these limitations, consumers expect fresh-cut products to maintain characteristics such as fresh-like appearance, taste, and flavor longer without the use of preservatives [9].

A natural alternative to maintain the quality characteristics, diminish the undesired physicochemical changes, and prolong the shelf-life of fresh-cut produce during the storage period, is the usage of edible barrier materials (i.e., polysaccharides, proteins, and lipids) [10,11]. Lipids constitute the most resistant edible coatings against moisture transfer owing to their hydrophobic character [12,13]. However, due to the presence of consumer concerns about lipid consumption and the creation of waxy mouth-feel [14], lipid-based coatings are not preferred as a base material for the coating of fresh-cut fruits and vegetables.

Alginate is a marine-origin polysaccharide extracted from brown algae [15,16]. It is classified as a GRAS (generally regarded as safe) substance by the US Food and Drug Administration (FDA), listed as an authorized food additive by the European Commission (EC), and used as an emulsifier, stabilizer, thickener, and gelling agent in the food sector [17,18]. Since alginate is a polyuronide, a natural ion exchanger, the addition of certain bivalent cations (e.g., Ca^2+^, Sr^2+^, Ba^2+^) into an alginate solution induces conformational changes such as the formation of the egg-box model [19] and leads to a gel formation through the bounding of bivalent ions between two chains of alginate and the formation of divalent salt bridges [20,21,22,23]. The immersion of an alginate film/coating into a calcium solution initiates two type of reactions: (i) insolubilization of the alginate film, which is induced by the diffusion of multivalent ions and the formation of linkage; (ii) the dissolution of alginate by the solution [24,25]. The dominancy of the dissolution process is suppressed by increasing the concentration of the bivalent ion [24]. Moreover, the application method of the bivalent ion has an impact on film thickness; for instance, the immersion of the alginate gel into a crosslinking solution leads to thinner film formation compared to the direct addition of the crosslinking agent into the alginate solution [26]. Alginate-based coatings and films crosslinked with Ca^2+^ preserve the quality characteristics and extend the shelf life of the food products by acting as a barrier to flavor volatiles and gases, impeding the loss of moisture, preventing microbial contamination and fat oxidation, maintaining textural stability, and preventing surface discoloration [27,28,29,30].

Recently, composite or multicomponent films have been designed as bilayers or emulsions to benefit from the complementary advantages of hydrophilic and hydrophobic compounds together [10,31]. Due to the requirement of four preparation stages (i.e., two casing and two drying stages), bilayer films have not been frequently focused on. On the other hand, there are numerous studies on the preparation of emulsion systems with hydrocolloidal components and dispersed lipid components such as vegetable oils, waxes, or fatty acids [31].

Water is a predominant component and constitutes 70–90% of the weight of fruits and vegetables [32]. The water contents of cantaloupe and strawberries were determined to be 92.8% w.b. (wet basis) and 90.7% w.b., respectively [33,34,35]. There are three states of water in food products: (i) free water (acts as a solvent or dispersing agent), (ii) adsorbed water (is held tightly), and (iii) water of hydration (bound chemically) [36]. The microbial stability, chemical and enzymatic reaction kinetics, textural properties, and physical stability of food are strongly related to the gain or loss of moisture in the food system and consequently end up with a reduction in shelf-life [14,37]. In other words, water loss, which is mainly caused by transpiration, is the major cause of food deterioration as a result of inducing an unappealing appearance (wilting, shriveling, etc.), textural alterations (softening, crispness, etc.), and a loss of nutritional quality and marketable weight [38,39]. Transpiration is a mass transport process, i.e., water vapor transfer from the food surface to the surrounding air [39]. When there is a water activity (a_w_) gradient between the food product and its environment, moisture transfer (transfer of liquid water and vapor) can occur from higher water activity (i.e., fruits) to lower water activity (i.e., packaging environment) until the thermodynamic equilibrium is reached [13,14,40,41]. Three phenomena occur during the mass transfer of water in foods. These are: (i) water transport within the product; (ii) water vapor transport between the surface of the product and the surrounding environment; and (iii) equilibrium between the water content of the food and the water content of the surrounding environment [42].

The rate of transpiration is directly proportional to the partial pressure gradient and transfer areas but inversely proportional to the surface resistances [39]. When viewed from this aspect, edible barriers such as coatings and films enhance the surface resistance of the produce. It is, therefore, necessary to understand the effect of an edible coating on water loss to select the most suitable coating formulations, conditions for packaging, and storage life.

Fruit manufacturers state that water accumulation at the bottom of the fresh-cut fruits’ packaging is a very important problem, decreases the value of the product, and discourages customers from buying. The fruit leakage can be observed especially in fresh-cut melon, watermelon, pineapple, etc. The objective of this study is to evaluate the efficacy of the previously optimized alginate-based coating formulation [43] based on the water barrier properties of coated fresh produce, i.e., strawberry and fresh-cut cantaloupe. To this end, coating uniformity was achieved with the addition of an extra immersion step into a calcium lactate solution initially before the coating step. To the best of the authors’ knowledge, this application method has not been utilized so far for the products that could not be uniformly coated with an alginate-based edible coating due to the high moisture content of their surface (hydrophilic surface, fresh-cut cantaloupe). In addition to this, the uniform gel-forming performance of the coating was also tested on a very hydrophobic surface (i.e., strawberry). Afterwards, the water loss, water activity, water vapor resistance characteristics of coated (with the new method, the extra immersion in calcium lactate solution) and uncoated products were measured and compared throughout the storage to identify whether the coating prevents the fruit from drying out. Furthermore, the differences in the mass losses of coated and uncoated products caused by a gradual decrease in relative humidity (RH) were monitored. Additionally, the water vapor transmission rate (WVTR) of stand-alone alginate films was measured for the assessment of barrier properties under ideal conditions.

## 2. Materials and Methods

### 2.1. Materials

Sodium alginate (Manugel GHB, FMC Biopolymer Co., Philadelphia, PA, USA), glycerol (Sigma–Aldrich Chemie GmbH, Steinheim, Germany), sunflower oil (Rewe Bio, Rewe Markt Gmbh, Köln, Germany), tween 80 (polyoxyethylenesorbitan monooleate) (Sigma–Aldrich Chemie GmbH, Steinheim, Germany), span 80 (sorbitan monooleate) (Sigma–Aldrich Chemie GmbH, Steinheim, Germany), calcium L-lactate hydrate (Sigma–Aldrich Chemie GmbH, Steinheim, Germany) were used in coating formulations.

Whole cantaloupe (*Cucumis melo* var. *cantalupensis*) and strawberries (*Fragaria* × *ananassa* D.) were purchased from Krohns Obst-Gemüse-Express (Berlin, Germany), Schweiger Obst-und Gemüsehandel (Freising, Germany), and a local market in Freising (Germany). Samples were transported directly to the laboratory.

### 2.2. Preparation of Food Sample

Fruits with no external defects were selected for the experiments. Experiments were conducted always at the same day of the fruit purchase and samples were stored at 4 °C room until used.

Cantaloupes were peeled, cut into two halves, the seeds were removed, and the remaining fruits were cut into pieces in accordance with the requirements of the experiments. Due to the large number of samples (in water loss (%) and water activity determination), or the easiness of capturing good stereomicroscope images (in thickness determination), cuboid pieces (14.1 ± 1.1 g, *n* = 20, ≈4 × 2 × 2 cm^3^) were prepared. For the experiments (i.e., water desorption as a means of mass loss during RH(%) decrease and water vapor resistance tests) in which special geometry was necessary for the experimental setup, ease of calculations and/or specified devices, cylinder pieces were cut as described in the relevant sections.

Strawberries were not hulled, cored or cut, but used as whole fruit. They were assigned to the experiments in accordance with the requirements of the associated experiments; e.g., due to the small sampling cups of the automated water sorption analyzer, strawberries with similar, small volumes were chosen. Pieces were transferred into a large food container with a lid and assigned to the treatments randomly.

### 2.3. Preparation of Edible Coating

Coating solution was prepared according to a previous study by Senturk Parreidt, Schott, Schmid, and Müller [43]. Sodium alginate (1.25%, *w*/*w*) was dissolved in distilled water with continuous stirring (magnetic stirrer (500 rpm)) at 70 °C until complete dissolution and a clear solution was achieved. Two percent glycerol (*w*/*w*) was added into the formulation to increase coating flexibility. Surface active agents (1% span 80 (*w*/*w*) and 0.2% tween 80 (*w*/*w*)) were incorporated into the coating formulation to improve adhesion of the coating on the product. Sunflower oil (0.2%, *w*/*w*) was added as a lipid source to increase water barrier characteristics. The concentration of oil was kept low since target food materials are fruits and consumer acceptance against high oil-incorporated fruit may be low. Subsequent to the continuous stirring with a magnetic stirrer (IKA Werke GmbH & Co. KG, Staufen, Germany) to achieve the dissolution of ingredients, mixtures were homogenized and emulsified using an ultra-turrax homogenizer (Miccra D-8, ART modern Labortechnik GmbH, Müllheim, Germany) at 10,500 min^−1^ for 5 min. The solutions were degassed at room temperature (~20 °C) in an ultrasonic bath (Transsonic 460/H, Carl Roth GmbH Co. KG, Karlsruhe, Germany) at a frequency of 35 kHz for another 5 min.

To induce the gelling mechanism and crosslinking reaction, 2% calcium lactate was dispersed in distilled water.

### 2.4. Coating Application

#### 2.4.1. Conventional Alginate Coating

Coating applications were performed according to the process parameters (i.e., dipping and draining periods) stated by Senturk Parreidt, Schmid, and Müller [28] and illustrated in Figure 1a. Fruits were immersed into the alginate solution for 2 min and the excessive hydrocolloid solution was drained for 1 min. Afterwards, coated samples were immersed into a 2% calcium lactate solution for 2 min to achieve gel formation, and the residual solution was allowed to drip off for 1 min. Subsequent to the coating process, strawberries were dried at room temperature (~20 °C) for 35 min. However, coated fresh-cut cantaloupe pieces were not allowed to dry to keep their juicy surface image. Preliminary sensorial evaluations showed that consumers preferred to see watery, juicy fresh-cut cantaloupe samples instead of a cut surface covered with a relatively dry gel formation.

#### 2.4.2. Novel Alginate Coating

To achieve a uniform coating layer, additional dipping and draining processes were included initially before the conventional coating process (Figure 1b). Fruits were dipped in a calcium lactate solution for 2 min and the residual solution was allowed to drip off for 1 min. Afterwards, the same coating procedure, which was described in the conventional alginate coating section, was applied.

It is important to note that a novel coating method was used to test water barrier characteristics.

### 2.5. Coating Uniformity and Thickness

Coating uniformity, adherence to the fruit surface, and coating thickness were monitored using a stereomicrograph (Leica MZ16, Leica Mikrosysteme Vertrieb GmbH, Bensheim, Germany) by randomly taking five measurements at different points of the cross section of 10 samples (*n* = 10 × 5). The coating uniformity of samples, which were coated with conventional and novel alginate coating methods, and coating thicknesses were determined and compared.

### 2.6. Storage Conditions

Wills, McGlasson, Graham, and Joyce [2] and Jiang and Fu [44] reported that a RH of 90% is often considered as the best compromise condition for the storage of fruit. In the present study, a 90% RH level was chosen as the storage condition. The storage temperature was set to 10 °C, the minimum operating temperature for 90% RH storage in a constant climate chamber (APT.Line KBF, WTB Binder Labortechnik GmbH, Tuttlingen, Germany).

### 2.7. Weight Loss (%)

Fruits after coating and dripping were packed in aPET (amorphous polyethylene terephthalate) transparent boxes with an attached lid (250 mL, 117.25 mm × 112.5 mm × 41 mm, thickness = 250 µm, item no. 80,441, Meier Verpackungen GmbH, Hohenems, Austria). Coated and uncoated fresh-cut cantaloupe and strawberries were kept on shelves inside a constant climate chamber at 10 °C and 90% RH over a storage period of 14 days. Weighing was done using an analytical laboratory scale (Sartorius Lab Instruments GmbH & Co. KG, Goettingen, Germany) with 0.001 g sensitivity. To neutralize the static electricity at the work area, which originated from plastic packaging, and to increase the accuracy of the results, an ionizing blower (18 V, AC, 2 W, Sartorius Lab Instruments GmbH & Co. KG, Goettingen, Germany) was used during packaging and weighing.

Weight loss was determined in each package at different sampling dates by a percentage of weight loss (Equation (1)) with respect to day 0 (*weight_initial_*). Measurements were performed in four replicates (*n* = 4). Weight loss graphs were plotted with respect to time. Since a comparison was made using both coated and uncoated samples from the same batch, the contribution of the respiration rate to weight loss could be neglected.
(1)$$Weight\;loss\;\left(\%\right)=\frac{Weight_{initial}-Weight_{final}}{Weight_{initial}}\times 100$$


### 2.8. Water Desorption as a Means of Mass Loss During Relative Humidity Decrease

The percentage of the mass change of coated and uncoated food samples was measured as a function of time by means of an SPSx-1µ automated water sorption analyzer (ProUmid GmbH & Co. KG, Ulm, Germany). Subsequent to the removal of the thick rind (exocarp) and seeds, cantaloupe cylinders were prepared from fleshy mesocarp using a metal pastry cutter (diameter = 4.19 cm, height = 1.9 cm). In the case of strawberries, relatively similar sized cone-shaped samples were chosen. Samples were placed in individual aluminum weighing trays (diameter = 64 mm) on the sample carousel arranged in a circle and equilibrated at 80% RH and 10 °C. Afterwards, the relative humidity of the chamber was gradually decreased (5%) while the temperature was kept constant. Every next humidity decrease was induced when equilibrium was reached. Equilibrium was defined by less than 0.01% mass change per 75 min (equals 5 weighing cycles). The weight of the samples was recorded at 15 min intervals during 236 h with 0.001 mg sensitivity until 60% RH was reached.

### 2.9. Water Activity (a_w_)

Subsequent to the fruit coating (in the case of strawberry, also drying), fruits were packed in (i) closed, (ii) perforated, and (iii) open aPET transparent boxes. Perforation was provided by punching nine holes on the top surface of the packages using a sewing needle (diameter = 2 mm, Prym Consumer Europe GmbH, Stolberg, Germany).

The water activity of the samples was measured at 20 ± 0.2 °C using a dew point water activity meter (Aqualab 4TEV, METER Group AG, Munich, Germany) with the accuracy of ±0.003. The device was calibrated using two standards (i.e., 6.0 Molal NaCl in H_2_O (a_w_ = 0.760) and 0.5 Molal KCl in H_2_O (a_w_ = 0.984)). After calibration, samples were cut in the form of plastic sample cups and placed inside the measuring chamber.

### 2.10. Water Vapor Resistance (WVR)

The WVR of coated and uncoated strawberries and fresh-cut cantaloupe pieces was evaluated gravimetrically as described by Poverenov et al. [45] using Equation (2). Subsequent to the removal of the thick rind (exocarp) and seeds, cantaloupe cylinders were prepared from fleshy mesocarp using a metal pastry cutter (diameter = 4.19 cm, height = 1.9 cm). In the case of strawberries, relatively similar sized cone-shaped samples were chosen (diameter = 2.6 ± 0.3 cm, height = 3.0 ± 0.2 cm, *n* = 20). Water vapor resistance is defined as follows:
(2)$$WVR=\left[\frac{\left(a_{w}-\frac{\%RH}{100}\right)\times p_{WV}}{R\times T}\right]\times \left(\frac{A}{J}\right)$$
where *WVR* is s/cm, a_W_ is the water activity of strawberries or fresh-cut cantaloupe pieces, %*RH* is the relative humidity of the climatic chamber (=90%), *p_WV_* is the saturated water vapor pressure at 10 °C (9.21 mmHg [46]), *R* is the specific gas constant for water vapor (461.5 J kg^−1^ K^−1^ = 3461.544 mmHg cm^3^ g^−1^ K^−1^), *T* is the temperature of the climatic chamber (283.15 K), *A* is the surface area of the food products (38.8 cm^2^ for cantaloupe and 13.3 ± 2.0 cm^2^ for strawberry), and *J* is the slope of weight loss in food product versus storage time (g/s). The area of cantaloupe pieces was assumed as cylinders, while the shapes of strawberries were assumed as cones. *J* and a_w_ were measured using devices that were previously defined in Section 2.8 and Section 2.10.

Coated and uncoated pieces were placed in the middle of individual, plastic trays (diameter = 9 cm) and placed on shelves inside a constant climatic chamber (APT.Line KBF, WTB Binder Labortechnik GmbH, Tuttlingen, Germany) at 10 °C and 90% RH over a storage period of 14 days. The RH and temperature of the climatic chamber were also recorded over time via portable data logger to monitor the environmental conditions.

Control tests with uncoated fresh-cut cantaloupe and strawberries were performed to determine the resistance factor of the uncoated fruits to water vapor. Since comparison was made using both coated and uncoated samples from the same batch, the contribution of respiration rate to weight loss could be neglected.

### 2.11. Water Vapor Permeability (P) of Alginate Films

Aqueous alginate solutions were prepared with the same method and ingredients as previously described in Section 2.3. Different amounts of solutions (10, 15, 20, 30 g solution/plate) were cast on various sizes of glass and plastic Petri plates, and subsequently, a calcium lactate solution was sprayed on them. From that, films were obtained following the drying at 23 °C and 50% RH for 30 days in a climate chamber. Since RH was kept at a constant level in the chamber, films did not dry during measurements.

Water vapor transmission rate (WVTR) was measured with the gravimetric method according to DIN 53122-1 and using the modified ASTM method E 96–95 [47,48]. Cups were filled with distilled water, then stored in a climate chamber (Binder GmbH, Tuttlingen, Germany) at 23 °C and 50% RH. The initial weight of the cups and the weight during storage (each 1 h) was measured (Mettler H315; Mettler–Toledo GmbH, Gießen, Germany) until the weight gain stagnated. Three replicates of each of the seven specimens in different thicknesses (52–122 µm) were tested. The WVTR was calculated using the following Equation (3):
(3)$$WVTR=\frac{24}{\Delta t}\cdot \frac{\Delta m}{A}\cdot 10^{4}\;\left(\frac{g}{m^{2}\cdot d}\right)$$
where Δ*t* (h) is the time between two weight measurements of which Δ*m* is calculated, Δ*m* (g) is the weight difference of two successive weight measurements, and *A* (cm^2^) is the sample area. To make these values comparable for different film thicknesses (*d*), the WVTR is multiplied with the thickness of the sample (*Q*_100_):
(4)$$Q_{100}=WVTR\cdot d\;\left(\frac{g\cdot 100\;µm}{m^{2}\cdot d}\right)$$


As other researchers measured the *WVTR* at different partial pressure differences (Δ*p*), this information is further transformed into the permeability values *p*:
(5)$$P=\frac{Q_{100}}{\Delta p}\;\left(\frac{g\cdot 100\;µm}{m^{2}\cdot d\cdot \mathrm{hPa}}\right)$$


Strictly speaking, this transformation is not valid, as the law of Henry does not apply for such polar polymers for water vapor but is accepted as a simplification [49].

### 2.12. Statistical Evaluation

All experiments were performed in quadruplicate (*n* = 4). The mean and standard deviation were determined in Microsoft Excel 2010 (Microsoft Corp., Redmond, WA, USA). Graphics and statistical evaluations were performed using R 3.3.2 for Windows with the packages ggplot2 [50], grid [51], gridExtra [52], car [53], and lsr [54]. The types of statistical tests applied to the results were denoted individually in the relevant results and discussion sections.

## 3. Results

### 3.1. Coating Uniformity and Thickness

Figure 2a–f illustrates the stereomicroscope images of uncoated and coated (both conventional and novel coating method) products, which generally allows for a qualitative evaluation of coating uniformity. As can be seen in Figure 2b, the conventional coating method formed thicker gels especially on the edges/corners and thinner structures on the middle. This issue is most likely caused by a lack of the adhesion ability of the alginate solution on the highly hydrophilic cut surface. Figure 2e shows that the conventional coating method had better adhesion on the strawberry epicarp, thus, the coating spread more uniformly compared to the surface of the cantaloupe. However, as can be easily realized in Figure 2f, initial immersion into a calcium solution led to a gel formation that completely covers the low and high points (e.g., seed (achene) holes), forming a smoother gel surface.

Cross-sections of alginate-coated cantaloupe (Figure 2c) and strawberry images (Figure 2f) evidence the achieved coating uniformity by the novel coating method. The initial dipping into the calcium lactate solution generates a thin calcium lactate layer on the surface of the cut cantaloupe. In this way, alginate molecules can interact directly with calcium and form a gel on the surface. As is evident from the figures, the addition of an extra immersion step into a calcium solution led to thicker and more homogeneous gel formation. Average thicknesses of applied alginate coatings were measured as 0.187 ± 0.076 mm and 0.235 ± 0.077 mm for cantaloupe and strawberries, respectively.

### 3.2. Relative Weight Loss during Storage at 10 °C, 90% RH

Variations in water loss values of coated and uncoated products during 10 °C, 90% RH storage are presented in Figure 3. To eliminate the difference in the sample sizes, results were expressed as a percentage (%) of weight loss.

The pattern of water loss increase was similar for all experimental groups; the weight loss (%) of cantaloupe and strawberry samples increased with storage time. According to the conducted two-way ANOVA tests, coating application, storage time, and their interactions all had significant effects (*p* < 0.05) on the amount of water loss in both the cantaloupe and strawberry samples.

The coating process decreased the weight loss of the fresh-cut cantaloupes significantly compared to uncoated samples (Figure 3a). Post-hoc comparisons using the Tukey HSD (Honestly Significant Difference) test indicated that from day 0 to day 8, there were no significant differences between the samples in the same treatment group (*p* > 0.05).

In contrast to the cantaloupe application, two-way ANOVA tests revealed that the coating process significantly increased the water loss in the strawberry samples. Particularly after 10 days of storage, the difference in water loss increased drastically.

### 3.3. Water Desorption as a Means of Mass Loss During Relative Humidity Decrease

The relationship between the time and mass loss (%) of the samples obtained with and without coating application during 80%→60% RH is given in Figure 4. It is a similar experiment to those presented in the previous section. In theory, the sorption measurement device enables one to examine each sample piece individually, allows one to observe the effect of variable RH, and provides a comparison with the previous results.

The masses of the strawberry samples (independent of being coated or not) decreased linearly with time. However, the slope of the decreasing functions of the coated sample was smaller than the uncoated samples (m_coated_ < m_uncoated_ < 0). It is interesting to note that at the very beginning of the experiment (<20 h), coated strawberries showed a drastic decrease in mass loss followed by a relatively linear decrease, while uncoated samples had a linear decrease throughout the entire process.

According to Figure 4, there is, however, no distinguishable differences between coated and uncoated cantaloupe samples. On the contrary, it can be clearly seen from the graph that coated strawberries had higher mass loss compared to uncoated samples. This is also in agreement with Figure 3. To perform statistical evaluations, the curves presented in Figure 4 were assumed linear, and the average drying speed (|dm/dt|, %·h^−1^) was calculated using the boundary conditions (i.e., *t*_1_ = 0 h and *t*_2_ = 235 h) presented in Table 1. The Kruskal–Wallis rank–sum test and pairwise comparisons using the Wilcoxon rank–sum test revealed that there was no statistical difference between uncoated and coated cantaloupes; however, both coated and uncoated strawberry samples were significantly different (*p* < 0.05).

### 3.4. Water Activity (a_w_)

Throughout the duration of moisture transfer between a food and its environment, water activity as well as the moisture content of the product become a function of time [14]. Hence, the water activity values of coated and uncoated fresh-cut cantaloupe and strawberry samples, packed in closed, perforated, and open aPET transparent boxes were determined on the 1st, 5th, and 8th storage days (Figure 5).

For fresh-cut cantaloupes (Figure 5a), statistical evaluations (two-way ANOVA) revealed that the difference in packaging applications did not cause a significant change between a_w_ results (*p* > 0.05); however, storage time and the interaction of the type of packaging × storage time had a significant effect on a_w_ results (*p* < 0.05). The water activity values of cantaloupe samples on day 1 were significantly higher than day 5 and day 8. It can be concluded that the amount of water in the food product that can take part in chemical and physical reactions slightly decreased.

For strawberries (Figure 5b), results showed an increasing pattern as the storage period prolonged. Despite the drying of coated products, coating increased the available, unbound water amount. Two-way ANOVA results revealed that both the time, type of packaging, and their interaction had significant effects on a_w_ results (*p* < 0.05). Moreover, the samples stored in perforated packages showed a drastic increase.

### 3.5. Water Vapor Resistance (WVR) of Edible Coating

Table 2 compares the results obtained from the WVR analysis of coated/uncoated fresh-cut cantaloupe and strawberry samples. The WVR of the fruit surfaces decreased significantly after the second day of storage.

It is apparent from the table that the resistance effect of intact strawberry epicarp against water transfer was superior to other groups. However, coating application decreased the surface resistance of strawberry against water transfer significantly (*p* < 0.05). Additionally, the effect of storage time was also significant at the *p* = 0.05 level.

### 3.6. Water Vapor Permeability of Edible Films and Expected Weight Loss During Storage

The measured values of WVTR were 2131 ± 299 g/(m^2^·d), which equals a Q_100_ of 1635 ± 372 g·100 µm/(m^2^·d) at (average films thickness was 78 µm), and a permeability of 103 ± 24 g·100 µm/(m^2^·d·hPa) (at a humidity of 100% RH→50% RH, at 23 °C, which equals 28 hPa→14 hPa). The single measured values can be found in the Appendix A online. These values are comparable to those achieved by other researchers, who produced cast films [48,55,56].

The approximate expected weight loss of cantaloupe and strawberry samples based on the measured water vapor permeability of alginate films was calculated and is illustrated in Figure 6. It is important to note that the calculation was based on the measured coating thickness of alginate films on the samples (0.187 ± 0.076 mm and 0.235 ± 0.077 mm for cantaloupe and strawberries); the storage conditions, as in Section 3.2 (10 °C, 90% RH in the atmosphere, 0.99 a_w_ in the fruit, equals a partial pressure difference of 1 hPa), and storage time (up to 15 days).

## 4. Discussion

### 4.1. Coating Uniformity and Thickness

Continuity and a uniform film thickness are important assumptions and prerequisites for the reliability and comparison of the measured values during the determination of thickness-dependent characteristics, such as the barrier properties of edible coatings [57,58]. By possessing very hydrophilic or hydrophobic surfaces, both samples encounter different types of challenges in forming uniform coating. The hydrophilic characteristic of the cut surfaces of fruits such as fresh-cut apple or cantaloupe does not allow for good adhesion of the coating material on the surface through a simple dipping method [59]. Due to the dilution of the coating by the surface moisture, the affinity of the coating material to the cut surfaces of horticultures is limited, which can lead to an uneven, discontinuous coverage of the targeted surface [45,57]. On the other hand, leafy vegetables and fruits such as strawberries have rough surfaces with micro- and nanostructures made of unwettable wax crystals and exhibit strong hydrophobicity [60].

Initial dipping into the calcium lactate solution improved the coating method and led to uniform layer formation on both very hydrophilic and hydrophobic fruit surfaces. It enables the present work and future studies on alginate-based coating to evaluate the transport rate of molecular components more accurately.

The average thicknesses of applied alginate coatings are 0.187 ± 0.076 mm and 0.235 ± 0.077 mm for cantaloupe and strawberries, respectively. Pavlath and Orts [61] and Skurtys et al. [59] specified the thickness of the edible coating layer usually inferior to 0.3 mm and 0.25 mm, respectively, and stated that a coating with higher thicknesses is designated as a “sheet” [59]. According to that, the crosslinked gel formed on the surface of the produce was still within acceptable thickness limits.

In the present study, the alginate-based coating formed a relatively thicker gel on the fresh-cut cantaloupe surface (≈187 µm) compared to the thickness values reported in the literature. Tapia et al. [22] measured the alginate-based coating thickness as 136.81 ± 14.05 µm on papaya; Rojas-Graü et al. [62] obtained 132.45 ± 20.48 µm on fresh-cut apples. On the other hand, Sipahi et al. [63] found 180 ± 2.0 µm for 1 g/100 g alginate and 412 ± 0.9 µm for 2 g/100 g sodium alginate incorporated coatings on fresh-cut watermelon surface. Similarly, Narsaiah et al. [64] found that the thickness of the alginate films increased with increasing alginate concentration incorporated into the coating formulation (i.e., 122.11 µm, 188.75 µm, and 300.46 µm for 1%, 1.5%, and 2% alginate concentrations, respectively). Therefore, the formation of a thicker gel can be ascribed to the difference of the alginate-based formulations (i.e., presence and amount of ingredients) that can lead to different solution viscosities or the difference in coating methods.

### 4.2. Relative Weight Loss during Storage at 10 °C, 90% RH

Free water stimulates microbial deterioration, the physical splitting of commodities, browning, callusing, rooting, and the sprouting of horticultural products [2]. The moisture loss process is a function of time and temperature [36].

In the case of cantaloupe, the values were not well comparable to those that could be theoretically expected, as in Figure 6. However, the final value of strawberries after 12 days of storage was close to the value expected based on the water vapor permeability measurements (~13.5% weight loss after 15 days of storage). Whereas the weight loss could be expected to correlate linearly with storage time based on pure permeability assumptions, in reality, many other factors play an important role, which is described in the following sections.

Statistical comparison revealed that from day 0 to day 8 there was no significant differences between cantaloupe samples in the same treatment group (*p* > 0.05). However, in the following days, the amount of water loss increased significantly and rapidly. This drastic increase in water loss can be explained by the decay of fresh produce, which originates from mechanical injuries, microorganisms, fast respiration rates, etc., and increases with longer storage periods. The increased deterioration also affects the structure of the produce and leads to accelerated water loss.

The water loss phenomenon of the wounded tissue was explained by Wills et al. [2] as follows: Wounding causes high concentration of solute release from the tissue. The high osmotic property of the solutes attracts water vapor and leads to droplet formation. Under high RH conditions (e.g., in package), the volume of the formed droplets continues to grow and continues to extract osmotica from the plant tissue.

Although alginate is a hydrophilic polymer and does not target the control of water vapor migration due to its hydrophilic nature, results showed that water loss in cantaloupe samples was significantly decreased. These phenomena can be explained by two mechanisms: (i) Fresh-cut melon is a porous product with 0.133 ± 0.006 real porosity (%) values [65]. The bulk movement of gas transfer (including water vapor, oxygen, and carbon dioxide) is controlled by small pores called stomate [2]. The pores might be plugged by coating, and in this way, the tissue structure on the surface might be modified [2]. (ii) The continuous alginate film might be considered a means of sacrificing moisture agent; i.e., moisture evaporates from the film instead of the fresh-cut surface of the cantaloupe [66,67,68,69].

In contrast to the cantaloupe application, water loss in strawberry samples was increased with coating application. The pronounced negative impact can be explained by the surface characteristics of the strawberry fruits. Leaves of higher plants and fruits are covered by cuticle, namely, an extracellular membrane that is composed of polymeric cutin matrix and soluble cuticular waxes [70,71]. Surface waxes, which can be divided in epicuticular (on the surface of cutin matrix) and intra-cuticular (embedded within the polymer framework) waxes, constitute the protective outer covering and establish a barrier against water loss [39,70,71,72]. The water-based coating might impair the aforementioned surface structure and also result in a decrease of WVR values.

### 4.3. Water Desorption as a Means of Mass Loss During Relative Humidity Decrease

At the very beginning of the experiment (<20 h), coated strawberries showed a drastic decrease in mass loss followed by a relatively linear decrease, while uncoated samples had a linear decrease throughout the entire process. This might show the drying process of the coating and followed by the water loss of the product, which was similar linearity with the uncoated samples. This effect could not be observed in coated cantaloupe samples, possibly due to the very fast water transport inside the sample, i.e., from the product to the coating gel, which kept the coating hydrated.

The permeation rate is equal to the driving force (i.e., pressure difference, Δ*p*) divided by the resistance (surface of the product with or without coating) [11]. Although the coating application increased the resistance of the surface, this effect might be very small compared to the big pressure differences between the relatively bigger chamber volume and very small surface area of the piece (Δ*p* between *p*_surrounding_ and *p*_fruit surface_). Therefore, coating did not influence the water loss results of each small piece used in the present experiment. On the other hand, coating decreased the water loss (%) amount of coated fresh-cut cantaloupe pieces packaged in small aPET trays in the previous section (Section 3.3). Unlike the experimental conditions in the vapor sorption analyzer, many coated pieces were packed in small packages with little air space. Due to the small driving force between little air space in the packages and increased resistance formed on many coated pieces, the coating influenced the water loss results.

It is a very interesting observation that the RH of the surrounding atmosphere (in other words partial pressure difference; a_w food_ − RH_surrounding_) did not influence the water loss amount of the samples within 80%→60% RH. This incident also indicates that the limiting factor to water loss may be the water transport, which takes place inside of the fruit.

In conclusion, the current experimental setup of the test might be revised in order to answer emerging scientific questions.

### 4.4. Water Activity (a_w_)

The migration of moisture is controlled mainly by the water activity (in other words, partial pressure), not by the water content, while the hydration of the components tends to achieve a balance in their water activity, not in their water content [73,74,75]. It is a well-known fact that most of the water in fresh or wet food exerts a vapor pressure that is very close to that of pure water [76]. Water activity values of fresh foods are 1.00–0.95, which makes them susceptible to spoilage from some yeasts, Gram-negative rods, and bacterial spores [77]. The water activity values of cantaloupe were determined as 0.95–0.993 [65,78,79,80]. Water activity of fresh strawberries were determined previously in the literature as 0.98–0.99 [34,35,81]. In the present study, the initial a_w_ values of the products were consistent with these previous findings (Figure 5).

The sorption isotherm (i.e., shows the relationship between the water content and water activity of a sample) of a food product with high water content had a J-shaped curve (Type III isotherm pattern) and at high water activity levels, where water is bounded due to the macro-capillary forces, and water content increases very rapidly with water activity [32,82,83,84,85]. The sorption isotherm of high water content products indicates that even a drastic decrease in water content does not cause a significant decrease in water activity. Hence, it was expected that water activity results would not be affected markedly during the storage in 90% RH.

The samples stored in perforated packages showed a drastic increase. The low water vapor transmission rate of the aPET trays combined with the out-coming water of the edible coating and the transpiration of the strawberries might cause a saturation of the package atmosphere and cause condensation inside the package with a very high a_W_ value of the strawberries (>0.99). Fishman et al. [86] reported that perforations (2 mm, 4 holes) on the package affected the oxygen concentrations to a much greater extent than the RH% of the inside atmosphere. Using the same calculation method, Müller and Gibis [87] demonstrated a calculation example for water loss in zucchini through the pinholes. According to the presented method, the water vapor flux through the nine pinholes could be calculated as ≈0.09 g/day. Therefore, even though the number of perforation holes was increased to *n* = 9, we can argue that they did not aid in decreasing the relative humidity inside the packages and led to elevated a_w_ values.

### 4.5. Water Vapor Resistance (WVR) of Edible Coating

The underlying reason for the attenuated resistance of coated strawberries against water transfer would be the alteration of the skin structure by the coating. As described above in weight loss studies, coating might harm the naturally occurring protective waxes on the strawberry surface.

On the other hand, the WVR results of coated and uncoated cantaloupe samples required more detailed evaluation. Once again, coated cantaloupe samples had a significantly lower amount of water loss compared to uncoated samples, and thus, they confirmed the results found previously in Section 3.2. Notwithstanding decreased water loss results, no significant differences were found between the calculated WVR values of coated and uncoated cantaloupe samples (*p* > 0.01).

In the present study, the WVR of coated cantaloupe and strawberry samples was determined as 6.77 ± 0.33 s/cm and 6.28 ± 0.76, respectively. Similarly, Poverenov et al. [45] determined the WVR of uncoated melon pieces as 7 s/cm and alginate-coated melon pieces as 9 s/cm. Rojas-Graü et al. [62] reported that sunflower oil added to an alginate coating increased the WVR of apples significantly, since uncoated apples had 15.70 s/cm, while coated samples had 19.2 s/cm. However, the experiment was conducted in chambers equilibrated at 33% RH and at 25 °C. In the present study, the low WVR of alginate-coated fresh-cut cantaloupes could be attributed to the high permeability values of a polysaccharide film at high RH conditions. Vargas et al. [88] and Perdones et al. [89] stated that the high RH (90%) of the storage atmosphere led to highly plasticized film formation, and therefore, its barrier characteristics were greatly reduced with increasing water vapor permeability. In addition, Vargas et al. [90] stated that the coating application of chitosan with a simple dipping method did not have any significant effect on the WVR of the samples; coated and uncoated samples had 1.87 s/cm and 1.72 s/cm WVR, respectively, after 9 days of storage.

### 4.6. Water Vapor Permeability of Edible Films and Expected Weight Loss During Storage

The reducing rates of water loss by the packaging material depends on the permeability of the package to water vapor transfer [2]. Investigating the transfer of water vapor through the stand-alone film, without the food product, can provide a researcher a chance to observe the effect in relatively ideal conditions and not on a complex multi-component system such as fruit.

According to Figure 6, which shows the approximate expected weight loss of cantaloupe and strawberry samples based on the measured water vapor permeability of alginate films, the weight loss of cantaloupe will be higher than strawberry samples. This computation is valid for *t* ≤ 10 days storage.

## 5. Conclusions

In the present study, the alginate-based coating layer was applied to fruits to protect them from the surrounding medium. The water vapor barrier performance of the alginate-based coating was presented on two different coated food products and as stand-alone films. In this way, the findings provide additional information about the water barrier characteristics of alginate-based coating.

Additional immersion in a calcium lactate solution at the beginning of the coating process promotes gel formation on the fruit surface and the uniformity of the coating. The new coating process allows researchers to improve the adhesion of their designed alginate-based coatings and may be considered a promising aspect of increasing the effects of edible coatings on quality parameters.

Coating treatment efficiently reduced water loss in fresh-cut cantaloupe pieces. However, it promoted water loss in the strawberry samples. The alginate-based coating process together with an additional calcium dipping step shows a promising effect on hydrophilic cut surfaces, especially on porous food samples, but the effect on waxy surfaces can be investigated in more detail. Moreover, different experimental setups for the vapor sorption analyzer can be employed to help this examination and to aid researchers in understanding the occurred phenomena better. In further studies, the vacuum impregnation method can be used instead of a simple dipping method to increase the penetration of the film forming solution into the porous product and increase the WVR of the surface.

As argued in discussions, results indicate that the limiting factor of water loss may be the transport of water inside the fruits. Future studies can fruitfully explore this issue further and investigate the association between these transport mechanisms.

## Figures and Tables

**Figure 1 foods-08-00203-f001:**
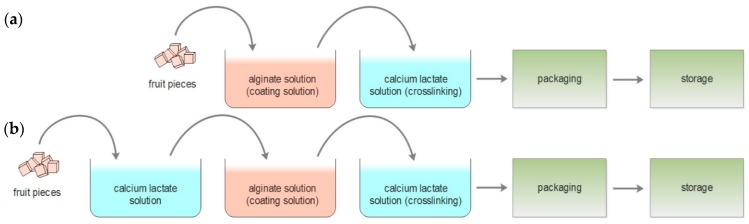
Schematic illustration of coating methods: (**a**) conventional dipping method to coat food products with alginate-based coatings; (**b**) novel dipping method with the incorporation of an extra immersion step into a calcium lactate solution.

**Figure 2 foods-08-00203-f002:**
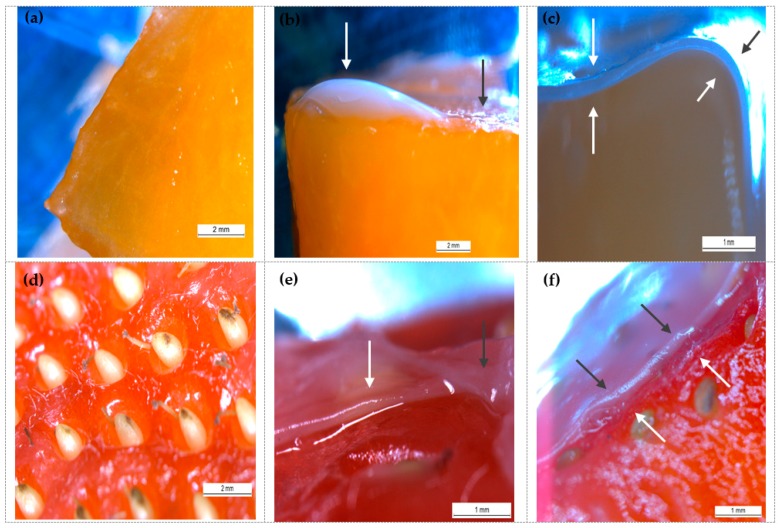
Stereomicroscope images of uncoated and coated fruits: (**a**) surface of uncoated cantaloupe; (**b**) cantaloupe surface coated with conventional method; (**c**) cantaloupe surface coated with novel method; (**d**) surface of uncoated strawberry; (**e**) strawberry surface coated with conventional method; (**f**) strawberry surface coated with novel method. Coating solution: 1.25% alginate + 2% glycerol + 0.2% sunflower oil + 1% span 80 + 0.2% tween 80. Gel formation was accomplished with a 2% calcium lactate solution. The color of the arrows does not designate anything and was only changed to increase visibility.

**Figure 3 foods-08-00203-f003:**
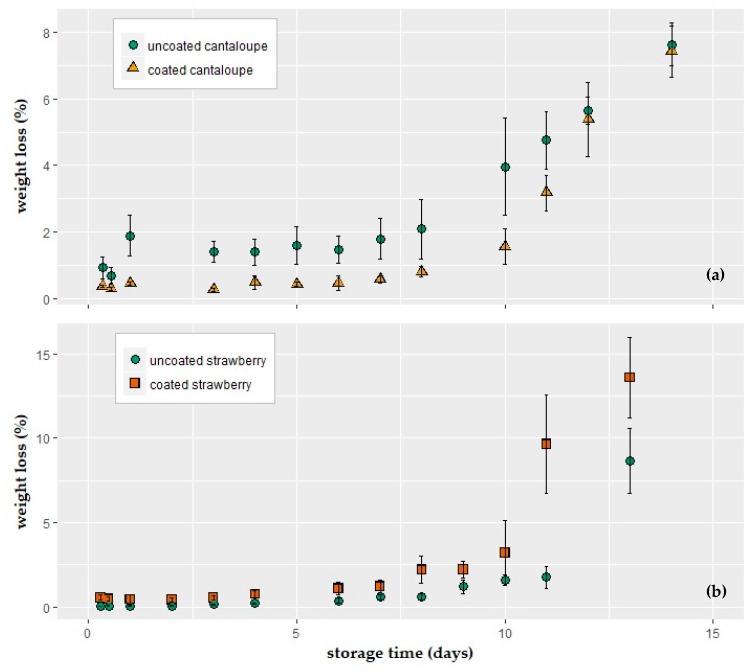
Weight loss of (**a**) fresh-cut cantaloupe; (**b**) strawberry fruits coated with an alginate-based solution (1.25% alginate + 2 glycerol + 0.2% sunflower oil + 1% span 80 + 0.2% tween 80 and 2% calcium lactate (crosslinking agent)) during storage at 10 °C and 90% relative humidity (RH).

**Figure 4 foods-08-00203-f004:**
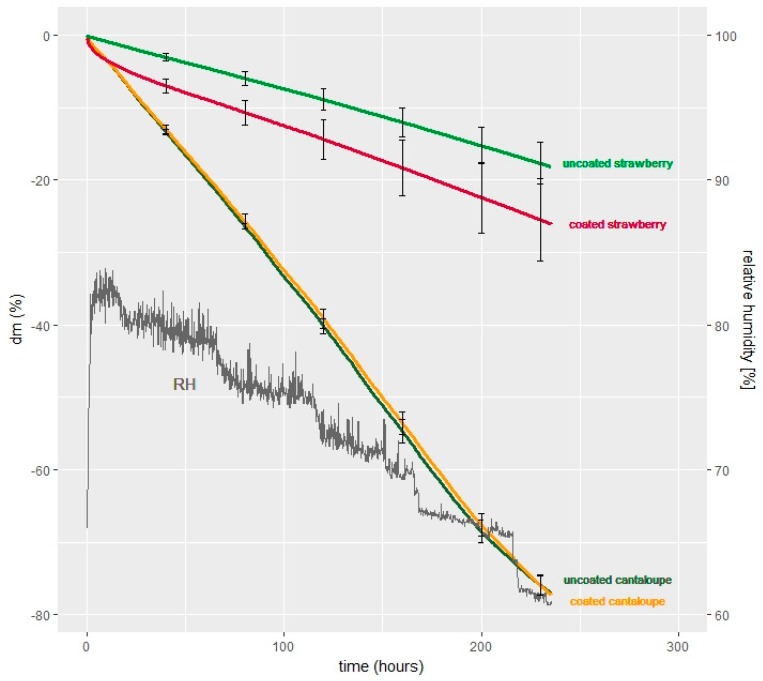
Mass loss (dm, %) of coated and uncoated strawberries and fresh-cut cantaloupes at 10 °C during gradual RH decrease (80%→60% RH) (*n* = 4). (Coating solution: 1.25% alginate + 2 glycerol + 0.2% sunflower oil + 1% span 80 + 0.2% tween 80 and 2% calcium lactate (crosslinking agent)).

**Figure 5 foods-08-00203-f005:**
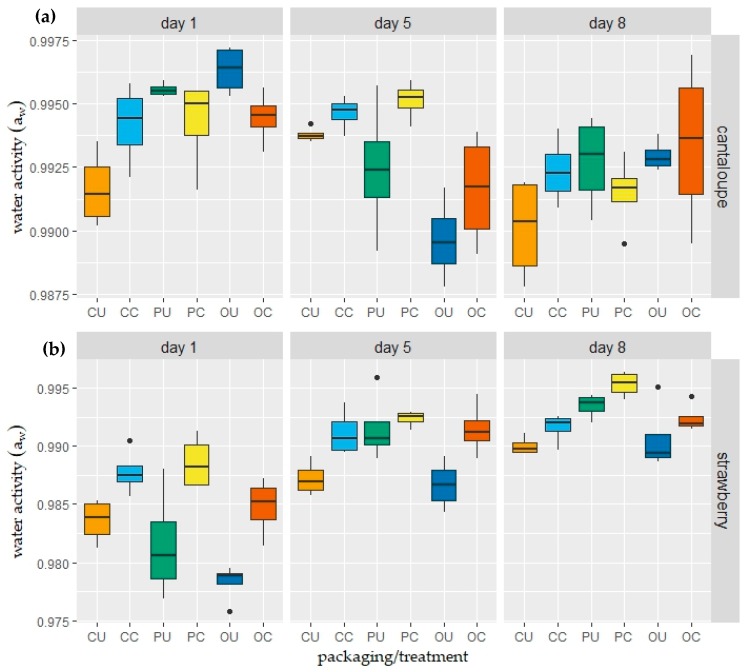
Box and whisker plots (median, min/max values, outliers) of water activity values of uncoated and coated (**a**) fresh-cut cantaloupe; (**b**) strawberry samples during storage at 10 °C and 90% RH. Alginate-based coating solution consists of 1.25% alginate + 2 glycerol + 0.2% sunflower oil + 1% span 80 + 0.2% tween 80 and 2% calcium lactate (crosslinking agent). CU: closed package–uncoated; CC: closed package–coated; PU: perforated package–uncoated; PC: perforated package–coated; OU: open package–uncoated; OC: open package–coated.

**Figure 6 foods-08-00203-f006:**
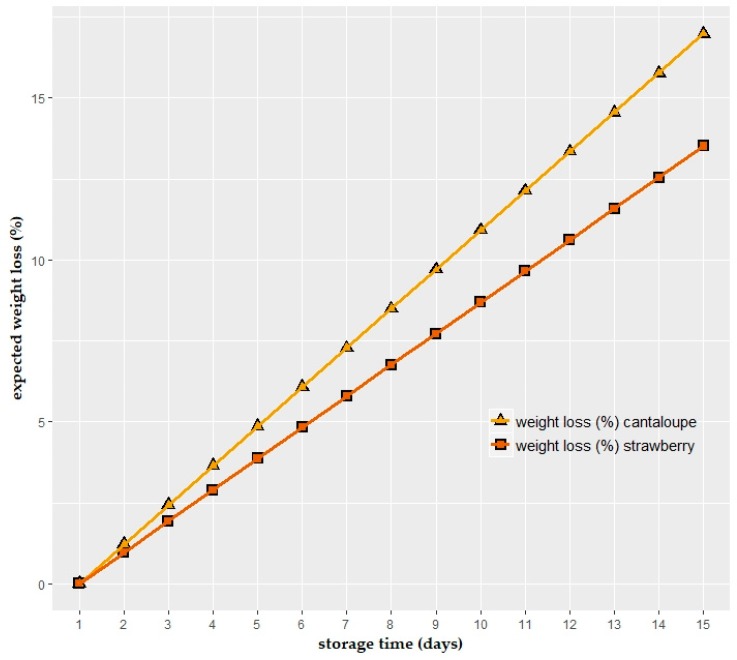
Expected weight loss (%) of cantaloupe and strawberry samples based on the measured water vapor permeability of alginate films produced in Petri dishes.

**Table 1 foods-08-00203-t001:** Average drying speed (|dm/dt|, %·h^−1^) (boundary conditions; *t*_1_ = 0 h and *t*_2_ = 235 h) of coated and uncoated strawberries and fresh-cut cantaloupes at 10 °C during gradual relative humidity (RH) decrease (80%→60% RH) (*n* = 4). (Coating solution: 1.25% alginate + 2 glycerol + 0.2% sunflower oil + 1% span 80 + 0.2% tween 80 and 2% calcium lactate (crosslinking agent)).

Drying Speed ^1,2^ (|dm/dt|) (%·h^−1^)			
Fresh-Cut Cantaloupe		Strawberry	
Uncoated	Coated	Uncoated	Coated
0.33 ± 0.01 _a_	0.33 ± 0.01 _a_	0.08 ± 0.01 _b_	0.11 ± 0.02 _c_

^1^ For average drying speed calculations, the curves (dm (%) versus time, Figure 5) were assumed linear. ^2^ For each group, similar small letters (subscript) in rows were not significantly different *p* ≤ 0.05.

**Table 2 foods-08-00203-t002:** Water vapor resistance (WVR) (s/cm) of uncoated and coated fresh-cut cantaloupe and strawberry samples throughout a 10-day storage period (*n* = 4). (Coating formulation: 1.25% alginate, 2% glycerol, 0.2% sunflower oil, 1% span 80, 0.2% tween 80).

Experiment Day	Fresh-Cut Cantaloupe ^1,2^		Strawberry ^1,2^	
	Uncoated	Coated	Uncoated	Coated
Day 2	7.05 ± 0.39 ^A^	6.77±0.33 ^A^	14.61 ± 3.36 ^C^_a_	6.28 ± 0.76 ^C^_b_
Day 4	2.79 ± 0.56 ^B^	2.86±0.84 ^B^	6.12 ± 1.91 ^D^_a_	4.64 ± 1.59 ^D^_b_
Day 6	2.40 ± 0.18 ^B^	2.66±0.23 ^B^	6.64 ± 2.07 ^D^_a_	3.92 ± 0.98 ^D^_b_
Day 8	2.43 ± 0.12 ^B^	2.93±0.40 ^B^	7.71 ± 1.32 ^D^_a_	3.44 ± 0.76 ^D^_b_
Day 10	2.31 ± 0.19 ^B^	2.82±0.12 ^B^	4.97 ± 0.30 ^D^_a_	3.89 ± 0.78 ^D^_b_

^1^ For each column, similar capital letters (superscript) were not significantly different at *p* ≤ 0.05 among days. For each treatment, similar small letters (subscript) in rows were not significantly different at *p* ≤ 0.05. ^2^ For cantaloupe, there was no significant difference between uncoated and coated groups.

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
