# Peer review of "The Development of a Uniform Alginate-Based Coating for Cantaloupe and Strawberries and the Characterization of Water Barrier Properties"

_foods, 2019, doi:10.3390/foods8060203_

Round 1
Reviewer 1 Report
The revised version of the paper now accounts for the requested revisions and corrections.
Please check for the symbol (?) in equation 3.
Author Response
Dear Reviewer,
Thank you for your helpful comments and for taking the time to point out options to improve our manuscript.
Sincerely yours,
Tugce Senturk Parreidt, Martina Lindner, Isabell Rothkopf, Markus Schmid and Kajetan Müller
Replies to the comments of Reviewer 1:
1. Please check for the symbol (?) in equation 3.
(Line 227) Equation was re-written.
Reviewer 2 Report
The authors did an overall revision in different parts of MS according to the editor and reviewers comments suggestions and improved the quality of MS.
Author Response
Dear Reviewer,
Thank you for your helpful comments and for taking the time to point out options to improve our manuscript.
Sincerely yours,
Tugce Senturk Parreidt, Martina Lindner, Isabell Rothkopf, Markus Schmid and Kajetan Müller
Reviewer 3 Report
The manuscript are improved and the changes gived an effective impact on the reading of results and on the advantages of the coating on the preservation of the tested products. In my opinion the work could be better after minor revision.
Minor revision:
Please check the typing in the equation 3;
I suggest to uniform in the text by write "cantaloupe " in the place which was writed only "melon".
Author Response
Dear Reviewer,
Thank you for your helpful comments and for taking the time to point out options to improve our manuscript.
Sincerely yours,
Tugce Senturk Parreidt, Martina Lindner, Isabell Rothkopf, Markus Schmid and Kajetan Müller
Replies to the comments of Reviewer 3:
1. Please check the typing in the equation 3.
(Line 227) Equation was re-written.
2. I suggest to uniform in the text by write "cantaloupe " in the place which was writed only "melon"
The word “melon” was changed into “cantaloupe” throughout the manuscript. However, only information taken from the literature was left as it was (melon).
This manuscript is a resubmission of an earlier submission. The following is a list of the peer review reports and author responses from that submission.
Round 1
Reviewer 1 Report
The idea it is good but unfortunately the obtained results did not support the hypothesis. In my opinion the only innovative result is the "novel coating method" that showed an uniform coatings (figure 2) but no improuvment in the shelf life of the tested producs was observed by using the new formulation respect to the uncoated product. Due to the ineffective action of new formulation i want to suggest to the authors to test different formulations of coating solutions in order to improve the shelf life moreover, not only the weight or water loss are interesting but could be interesting also other quality paramiters such as the texture of the fruits, the oxidative reacions, the color changes, etc.
Replies to the comments of Reviewer 1:
1. The idea it is good but unfortunately the obtained results did not support the hypothesis. In my opinion the only innovative result is the "novel coating method" that showed an uniform coatings (figure 2) but no improvement in the shelf life of the tested products was observed by using the new formulation respect to the uncoated product. Due to the ineffective action of new formulation i want to suggest to the authors to test different formulations of coating solutions in order to improve the shelf life moreover, not only the weight or water loss are interesting but could be interesting also other quality parameters such as the texture of the fruits, the oxidative reactions, the color changes, etc.
· (Line 105-108) An explanation of the main reason of the present study was included in “objective of the study” part:
“Fruit manufacturers state that water accumulation at the bottom of the fresh-cut fruits’ packaging is a very important problem, decreases the value of the product and discourages customers from buying. The fruit leakage can be observed especially in fresh-cut melon, watermelon, pineapple, etc.“
Water loss/water leakage is a very important problem that fresh-cut food industry encounters. Therefore our study focuses on this aspect. Moreover, strawberry were used to study the effects of the coating not only on a porous-high hydrophilic surface but also high hydrophobic waxy surface.
· The present manuscript does not propose a firm hypothesis that the designed coating formulation would increase the shelf life of the product. Instead, it underlines that the main focuses are uniform gel formation on the food product and evaluation of the effects on water barrier properties. Therefore, the aim of the study and the results were compatible with each other.
· In fact, the present study achieved its goals: Water leakage of fresh-cut cantaloupe samples were decreased significantly. It is a very interesting result because as being polysaccharide-based hydrocolloid, (as a rule of thumb) it was unexpected to function as water-barrier. Therefore, “water transport subject was evaluated with various measurement methods. The results also present interesting subjects for future studies, such as detailed evaluation of the water transport mechanisms in/through and on the porous fresh.cut food products; modification of the present WVR calculations for different types of products, detailed evaluation of the effects of coating on hydrophobic surfaces and waxy structures, etc.
· Reviewer’s suggestion/recommendation was very accurate, but it was performed in our previous study (Senturk Parreidt, T., Schmid, M., & Müller, K. (2018). Effect of dipping and vacuum impregnation coating techniques with alginate based coating on physical quality parameters of cantaloupe melon. Journal of food science, 83(4), 929-936). The effect of coating material on textural and color properties of the cantaloupe were presented.
Reviewer 2 Report
The paper is interesting, however I recommend a major revision. My comments are as follows:
Title is not relevant and need to be changed in order to indicate other research included.
Abstract should be improved. More information about obtained result are needed.
Why authors used so small amount of oil? (0.2%) For what reason oil was added?
More information about packaging should be provided.
Different volume of film-forming solution leads to different film thickness. More explanation about the reason why this method was used and film thickness should be added.
Did the films dry during 30 days? Please clarify
Description of obtained results is made not bad, however I recommend to add a table with the results of WVP. In general, this paper should be clearly divided for analyses about films and second about fruits. In this form, I see it rather as a little mess which does not help to understand easy this paper.
Replies to the comments of Reviewer 2:
1. Title is not relevant and need to be changed in order to indicate other research included.
· Title of the manuscript was changed:
“The development of a uniform alginate-based coating for cantaloupe and strawberries and the characterization of water barrier properties”.
2. Abstract should be improved. More information about obtained result are needed.
· Abstract was revised.
3. Why authors used so small amount of oil? (0.2%) For what reason oil was added?
· In introduction section (line 53-57), it is pointed out that: “Lipids constitute the most resistant edible coatings against moisture transfer owing to their hydrophobic character [12,13]. However due to consumer concerns about lipid consumption and the creation of waxy mouth-feel [14], lipid-based coatings are not preferred as a base material for the coating of fresh-cut fruits and vegetables.”
· In addition (Line 76-81) denoted that “Recently, composite or multicomponent films have been designed as bilayers or emulsions to benefit from the complementary advantages of hydrophilic and hydrophobic compounds together [10,31]. Due to the requirement of four preparation stages (i.e. two casing and two drying stages), bilayer films have not been frequently focused on. On the other hand, there are numerous studies on the preparation of emulsion systems with hydrocolloidal components and dispersed lipid components such as vegetable oils, waxes, or fatty acids [31].”
· (Line 157-160) The following explanation was added.
“0.2% sunflower oil (w/w) was added as a lipid source to increase water barrier characteristics. The concentration of oil was kept low since target food materials are fruits and consumer acceptance against high oil-incorporated fruit may be low.”
· The amount of 0.2% sunflower oil (w/w) were determined in the previous study (Senturk Parreidt et al. (2018)), which evaluated the effects of vegetable oils on alginate-based edible coatings in terms of coating stability and effective coating design for hydrophobic food surfaces.
4. More information about packaging should be provided.
· Additional information (i.e. Item No) was included (Line 204). This number enables a researcher to trace back the aPET packages.
5. Different volume of film-forming solution leads to different film thickness. More explanation about the reason why this method was used and film thickness should be added.
· You are fully right; different volumes lead to different film thicknesses. And different film thicknesses lead to different water vapor transmission rates. Therefore, the water vapor transmission rate [g/(m2*d)] is normalized to the thickness [g*100µm/(m2*d)] (equation 4). Like this, values become comparable. Film thicknesses and water vapor transmission rates will be supplied as supplementary materials.
6. Did the films dry during 30 days? Please clarify.
· The following explanation was added to Materials and Methods section (Line 268-269).
“Since RH was kept at a constant level in the chamber, films did not dry during measurements.”
7. Description of obtained results is made not bad, however I recommend to add a table with the results of WVP.
· Film thicknesses and water vapor transmission rates will be supplied as supplementary materials. The explanation was added into manuscript (Line 410):
“The single measured values can be found in the supplementary materials online.“
8. In general, this paper should be clearly divided for analyses about films and second about fruits. In this form, I see it rather as a little mess which does not help to understand easy this paper.
· To distinguish the analyses of „edible coating“ and „edible film“; the sections about alginate-based edible films were moved. Analysis of edible films (i.e. water Vapor Permeability of Alginate Films) are now in the following sections: Section 2.11 in Materials and Methods; Section 3.6 in Results; Section 4.6 in Discussion.
Reviewer 3 Report
This article presents interesting information about the water barrier properties of alginate coatings when applied to cantaloupes pieces and strawberry fruits. I may recommend some modifications to improve the impact among the specialists in the field.
I understand that the novelty of this article is the use of a preliminary calcium lactate dipping and, consequently, I assume that coatings essayed here are prepared with this novel methodology. However this is not specifically indicated in the text and it may be confusing for readers.
Along the text, a comprehensive study of water barrier properties of coated and uncoated specimen is carried out. But considering the new preparation method, I guess the comparison between samples initially treated and untreated with calcium lactate would fit better the aims of the paper. The benefits of the initial step should be highlighted. Accordingly, the introduction should focus on the previous studies covering the effect of ions in the preparation of alginate coatings rather than on very general issues such as the positive effect on coatings on the preservation of fruits and the water transport mechanisms. Authors already have a number of papers addressing these subjects.
A justification of the particular selection of cantaloupe and strawberry fruits should be provided. Differences in surface hydrophobicity are mentioned in some paragraphs but the motivations of the study and models selected should be introduced at the presentation of the work.
I also recommend an exhaustive grammatical revision, particularly in the Introduction and in the Materials and Methods sections.
Replies to the comments of Reviewer 3:
This article presents interesting information about the water barrier properties of alginate coatings when applied to cantaloupes pieces and strawberry fruits. I may recommend some modifications to improve the impact among the specialists in the field.
1. I understand that the novelty of this article is the use of a preliminary calcium lactate dipping and, consequently, I assume that coatings essayed here are prepared with this novel methodology. However this is not specifically indicated in the text and it may be confusing for readers.
· The following explanations were added to the new version of the manuscript;
(Line 116-119) “Afterwards, the water loss, water activity, water vapor resistance characteristics of coated (with the new method, the extra immersion in calcium lactate solution) and uncoated products were measured and compared throughout the storage to identify whether the coating prevents the fruit from drying out.“
(Line 188) “It is important to note that a novel coating method was used to test water barrier characteristics”.
2. Along the text, a comprehensive study of water barrier properties of coated and uncoated specimen is carried out. But considering the new preparation method, I guess the comparison between samples initially treated and untreated with calcium lactate would fit better the aims of the paper. The benefits of the initial step should be highlighted. Accordingly, the introduction should focus on the previous studies covering the effect of ions in the preparation of alginate coatings rather than on very general issues such as the positive effect on coatings on the preservation of fruits and the water transport mechanisms. Authors already have a number of papers addressing these subjects.
· Indeed, the comparison of both methods would be an interesting question. However, it was not the focus of the research. In the present manuscript, the primary focuses were defined as evaluating the effect of alginate-base edible coating on water loss process subsequent to the achievement of an uniform layer on the fruit surface.
· The benefits of the initial dipping in calcium lactate solution was highlighted once more in Discussion Section 4.1 as follows:
(Line 433-436) “Initial dipping into the calcium lactate solution improved the coating method and led to uniform layer formation on both very hydrophilic and hydrophobic fruit surfaces. It enables the present work and future studies on alginate-based coating to evaluate the transport rate of molecular components more accurately.”
· As recommended by the reviewer, the paragraph (Line 51-57) that gives general information about edible packaging were shortened. However, information about lipid-based edible coatings/films were not erased due to being particularly asked by another reviewer during the first revision process.
· Gelling mechanism and crosslinking effects of bivalent ions were explained in Introduction (Line 61-72) as follows:
“Since alginate is a polyuronide, a natural ion exchanger, the addition of certain bivalent cations (e.g. Ca2+, Sr2+, Ba2+) into an alginate solution induces conformational changes such as the formation of the egg-box model [19] and leads to a gel formation through the bounding of bivalent ions between two chains of alginate and the formation of divalent salt bridges [20-23]. The immersion of an alginate film/coating into a calcium solution initiates two type of reactions: (i) insolubilization of the alginate film, which is induced by the diffusion of multivalent ions and the formation of linkage; (ii) the dissolution of alginate by the solution [24,25]. The dominancy of dissolution process is suppressed by increasing the concentration of the bivalent ion [24]. Moreover, the application method of the bivalent ion has an impact on film thickness; for instance, the immersion of the alginate gel into a crosslinking solution leads to thinner film formation compared to the direct addition of the crosslinking agent into the alginate solution [26].”
· The paragraphs that gives general information about water transport mechanisms were not shortened due to referencing this information especially in water activity and water desorption sections.
3. A justification of the particular selection of cantaloupe and strawberry fruits should be provided. Differences in surface hydrophobicity are mentioned in some paragraphs but the motivations of the study and models selected should be introduced at the presentation of the work.
· The following explanations were added into manuscript.
(Line 105-108) “Fruit manufacturers state that water accumulation at the bottom of the fresh-cut fruits’ packaging is a very important problem, decreases the value of the product and discourages customers from buying. The fruit leakage can be observed especially in fresh-cut melon, watermelon, pineapple, etc.“
(Line 112-116) “To the best of the authors’ knowledge, this application method has not been utilized so far for the products that could not be uniformly coated with an alginate-based edible coating due to their high moisture content of the surface (hydrophilic surface, fresh-cut cantaloupe). In addition to that, the uniform gel-forming performance of the coating was also tested on a very hydrophobic surface (strawberry).”
4. I also recommend an exhaustive grammatical revision, particularly in the Introduction and in the Materials and Methods sections.
Grammatical or spelling error were corrected by an English language editing service.